# Histo-Blood Group Antigen-Producing Bacterial Cocktail Reduces Rotavirus A, B, and C Infection and Disease in Gnotobiotic Piglets

**DOI:** 10.3390/v16050660

**Published:** 2024-04-24

**Authors:** Sergei A. Raev, Maryssa K. Kick, Maria Chellis, Joshua O. Amimo, Linda J. Saif, Anastasia N. Vlasova

**Affiliations:** 1Center for Food Animal Health, Department of Animal Sciences, College of Food, Agricultural and Environmental Sciences, The Ohio State University, Wooster, OH 44691, USA; raev.1@osu.edu (S.A.R.); kick.28@osu.edu (M.K.K.); chellis.15@osu.edu (M.C.); saif.2@osu.edu (L.J.S.); 2GIVAX Inc., Philadelphia, PA 19106, USA; jamimo@givax.bio

**Keywords:** probiotics, rotavirus infection, histo-blood group antigens, glycans, diarrhea, shedding

## Abstract

The suboptimal performance of rotavirus (RV) vaccines in developing countries and in animals necessitates further research on the development of novel therapeutics and control strategies. To initiate infection, RV interacts with cell-surface *O*-glycans, including histo-blood group antigens (HBGAs). We have previously demonstrated that certain non-pathogenic bacteria express HBGA^-^ like substances (HBGA^+^) capable of binding RV particles in vitro. We hypothesized that HBGA^+^ bacteria can bind RV particles in the gut lumen protecting against RV species A (RVA), B (RVB), and C (RVC) infection in vivo. In this study, germ-free piglets were colonized with HBGA^+^ or HBGA^-^ bacterial cocktail and infected with RVA/RVB/RVC of different genotypes. Diarrhea severity, virus shedding, immunoglobulin A (IgA) Ab titers, and cytokine levels were evaluated. Overall, colonization with HBGA^+^ bacteria resulted in reduced diarrhea severity and virus shedding compared to the HBGA^-^ bacteria. Consistent with our hypothesis, the reduced severity of RV disease and infection was not associated with significant alterations in immune responses. Additionally, colonization with HBGA^+^ bacteria conferred beneficial effects irrespective of the piglet HBGA phenotype. These findings are the first experimental evidence that probiotic performance in vivo can be improved by including HBGA^+^ bacteria, providing decoy epitopes for broader/more consistent protection against diverse RVs.

## 1. Introduction

Rotavirus (RV) is the major causative agent of acute gastroenteritis and is associated with an increased risk of secondary bacterial infections in children and young animals globally [1]. In children younger than 5 years of age, severe RV-induced diarrhea may lead to hospitalization and even death [2,3]. RV mainly targets the mature terminally differentiated intestinal epithelial cells (IECs), primarily of the ileum and jejunum [4,5]. Among a variety of RV receptors, cellular glycans have been shown to play a major role as attachment sites [5]. Specifically, histo-blood group antigens (HBGAs), including the antigens of the ABO blood group system, have been shown to play a critical role in determining RV species/genotype-specific binding and disease [5,6,7,8]. O and A but not B have been described for pigs (AO system) [9,10]. This is determined by the presence of only two alleles, *A* and *O*, in the porcine *ABO* gene [11,12], resulting in the existence of four phenotypes: A, A^weak^, O, and “H^-^A^-^” [10]. Based on reactivity with “anti-A” and “anti-H” antibodies, pigs can be H^−^A+ (A phenotype), H^+^A^+^ (A^weak^ phenotype), H^+^A^−^ (O phenotype), and H^−^A^-^ (H^−^A^-^ phenotype) [12]. Our previous studies have demonstrated that the piglet HBGA phenotype affects RVA/RVC replication levels in vitro (in porcine ileal enteroids) [6,7]. This underscores the importance of considering the AO phenotype as an important factor influencing RV replication in vivo.

Before reaching its principal target, IECs, RV must penetrate the mucus layer, which protects IECs against enteric pathogens, including RV [13], and provides a niche for intestinal commensals [14]. There is growing evidence that several members of non-pathogenic bacteria produce glycans recognized by human HBGA-specific monoclonal antibodies [15,16,17,18,19]. Thus, while cellular HBGAs aid RV attachment, bacterial HBGAs might act as decoy epitopes, preventing RV attachment to the IECs. Our recent study has demonstrated the ability of some Gram-positive and Gram-negative non-pathogenic bacteria to express a variety of HBGAs and bind RV of different species (RVA/RVC) and genotypes in a genotype-specific manner [16,20], suggesting the potential role of these bacteria as decoy receptors for RV (Appendix A). However, the impact of these HBGA-expressing bacteria on RV infection and disease in vivo remains unknown.

While probiotic supplementation is generally beneficial in terms of the overall performance of livestock animals [21], feed conversion efficiency, and in reducing post-weaning diarrhea in pigs [21,22], it does not always meet producer expectations mostly due to the inconsistent outcomes [23,24,25,26]. This is likely due to variable dosages and types of probiotics used, animal diet, and age. While most studies on the impact of probiotics demonstrate immune-mediated decreases in viral shedding and clinical disease severity [20,27,28,29], the data on the role of direct bacteria–RV interactions are limited. We hypothesized that colonization of germ-free (GF) piglets with HBGA expressing (HBGA^+^) vs. non-expressing (HBGA^-^) bacteria would lead to decreased replication of RV, resulting in reduced diarrhea severity and virus shedding after virus inoculation, and that these beneficial effects will be independent of the probiotic-induced immunomodulation. Thus, the goal of this study was to evaluate the protective effects of HBGA^+^ vs. HBGA^-^ bacteria against RVA, RVB, and RVC infection in vivo.

## 2. Materials and Methods

### 2.1. Commensal Bacteria

We used two commensal facultative anaerobic bacteria (*L. brevis*, *S. bovis*) and four obligate anaerobes (*B. adolescentis*, *B. longum*, *B. thetaiotaomicron*, and *C. clostridioforme*) previously isolated from the gut of healthy pigs (kindly provided by Dr. David Francis, South Dakota State University, Brookings, SD, USA). An additional facultative anaerobe, *E. coli* G58 (kindly provided by Dr. Carlton Gyles, University of Guelph, Guelph, ON, Canada) was also included in this study. All strains were cultured under aerobic (*E. coli* G58) and anaerobic conditions (*S. bovis*, *B. thetaiotaomicron*, *B. adolescentis*, *L. brevis*, *C. clostridioforme*, *B. longum*); the latter were generated using the GasPakTM EZ Anaerobe Container System Sachets (BD, Franklin Lakes, NJ, USA). All bacteria were enumerated as described previously [30]. Selected media and growth conditions for preparing bacterial cultures were reported previously [16].

### 2.2. Rotaviruses

Intestinal contents of GF piglets containing rotavirus A (RVA): Wa G1P[8] [31], RV0084 G9P[13] [32], Gottfried G4P[6] [33], OSU G5P[7]; rotavirus B (RVB): Ohio [34] (non-typed); and rotavirus C (RVC): Cowden G1P[1] [35]; RV0104 G3P[18]; RV0143 G6P[5] [32], were used to orally inoculate piglets at a dose of 1 × 10^6^ fluorescent focus units (FFU).

### 2.3. Animal Experiments

All our animal experiments were approved by the Institutional Animal Care and Use Committee (IACUC) at Ohio State University (protocols #2009A0146, #2010A00000088). Near-term sows (Landrace × Yorkshire × Duroc crossbred) were purchased from the Ohio State University swine center facility/Shoup Brothers Farm LTD, Orville, OH, USA. GF piglets were derived by cesarean section and maintained as described previously [36]. On the 2nd day of life, rectal swabs were taken from all the piglets, and sterility was confirmed by culturing of rectal swabs in blood agar plates and thioglycolate broth culture. The presence of bacteria in the intestine vs. GF conditions has been shown to play a crucial role in nutrient absorption [37], immune system development [38], glycosylation profiles, and maintaining intestinal epithelial cell integrity [39], thus impacting immune responses to pathogens, including RV [40,41,42]. Therefore, to evaluate the anti-RV properties of the HBGA^+^ bacterial cocktail, instead of using non-colonized piglets, we used the HBGA^-^ bacterial cocktail as a control. Five- to seven-day-old GF piglets were supplemented for 5 consecutive days with one of the commensal bacteria cocktails (1 × 10^5^ colony-forming units, CFU, of each strain per pig) containing (1) HBGA-expressing bacteria (HBGA^+^): *Escherichia coli G-58*, *Bifidobacterium adolescentis*, *Bacteroides thetaiotaomicron*, *Streptococcus bovis*, and *Clostridium clostridioforme*; (2) HBGA-non-expressing bacteria (HBGA^-^): *Lactobacillus brevis* and *Bifidobacterium longum*. The HBGA expression profiles of the bacterial strains used in this study were evaluated in our previous study [16]. On day 5 of supplementation, rectal swabs were collected for enumeration of fecal bacterial shedding [30] in the colonized pigs. Additionally, the presence of each bacterial strain was confirmed using PCR, as described previously [43] (primers [44,45,46,47,48,49] are listed in Appendix A). On day 5 of supplementation, all piglets were inoculated with individual RVA/RVB/RVC strains at a dose of 1 × 10^6^ FFU/piglet. After the RV challenge, rectal swabs were collected daily to assess RV shedding and diarrhea severity, as previously described [50]. Blood samples were collected on days 0, 3, 7, and 11 post-infection (dpi) to evaluate the IgA Ab titers and canonical innate and pro-inflammatory cytokine responses to RV infection. An innate immune response early-response cytokine IFN-α was evaluated at dpi 0 and dpi 3, while TNF-α, IL-10, and IL-22 were assessed at dpi 0 and dpi 11 to capture the late phase of the immune response [51,52,53]. All piglets were euthanized at dpi 11, and small and large intestinal contents (SIC and LIC) were collected, resuspended at a 1:1 ratio in MEM with a protease inhibitor cocktail containing 250 µg/mL of trypsin inhibitor and 50 µg/mL of leupeptin (Sigma, Saint Louis, MO, USA), and stored at −70 °C to evaluate the local IgA response. Ileum sections were collected to determine porcine HBGA phenotype.

### 2.4. Rotavirus Fecal Shedding

Cell culture immunofluorescence (CCIF) assay was used to quantify RVA as previously described [54]. The final RV titers were calculated and expressed as the reciprocal of the highest dilution showing positive fluorescing cells. To detect/quantify RVC and RVB, real-time RT-PCR was used as previously described [55,56] (primers are listed in Appendix A).

### 2.5. Rotavirus A-Specific Antibody (Ab) ELISA Assay

Cell-culture-adapted RVA OSU G5P[7] and Wa G1P[8] strains were used to inoculate MA-104 cells, as described previously [57]. Infected cells were frozen/thawed 3 times, and after centrifugation, the supernatant was used as an antigen for IgA Ab ELISA (mock-infected MA-104 cells were used as a control). RV IgA ELISA was performed as described previously [58]. The RVA-specific IgA Ab titers were expressed as the reciprocal of the highest dilution that had a corrected optical density (OD)_450_ value (sample OD_450_ in the RVA antigen-coated well minus sample OD_450_ in the mock antigen-coated well) greater than the cut-off value (the mean + three standard deviations of negative control samples).

### 2.6. RVC ELISA

Ninety-six-well plates (Nunc Maxisorp, Thermo Scientific Pierce, Rockford, IL, USA) were coated with lysates (normalized for total protein content) of High-Five cells (Mock) or High-Five cells infected with the recombinant baculovirus containing the VP6 gene of RVC G1P[1] diluted 1:50 in carbonate–bicarbonate (coating) buffer (pH 9.6). After overnight incubation at 4 °C, the plates were rinsed twice with PBS containing 0.05% Tween-20 (PBST) and blocked with PBST containing 2% non-fat dry milk, and then incubated at 37 °C for 1 h. After rinsing the plates 5 times with PBST, seven 3-fold dilutions of serum samples (starting at 1:5) were added to both Mock/VP6-coated plates and incubated at 37 °C for 1 h. The plates were rinsed 5 times with PBST, and the secondary antibody, horseradish peroxidase-conjugated goat anti-porcine IgA antibody (Bio-Rad, Hercules, CA, USA), was added and incubated at 37 °C for 1 h. Then, the plates were rinsed 5 times with PBST, developed with TMB 2-Component Microwell Peroxidase Substrate Kit, and stopped with TMB Stop Solution (both from SeraCare Life Sciences Inc. Milford, MA, USA), and the OD values were read at 450 nm using SoftMax Pro 7.1 (Molecular Devices, LLC., San Jose, CA, USA). The antibody titers were determined as described previously [57].

### 2.7. Cytokine ELISA

Porcine TNF-α, INF-α, IL-10, and IL-22 ELISA kits were used as described in the manufacturer’s recommendations (Thermo Scientific Pierce, Rockford, IL, USA).

### 2.8. HBGA Immunohistochemistry

To determine porcine HBGA phenotype, formalin-fixed ileal sections [59] were stained with HBGA-A- or -H-specific mouse monoclonal antibodies (mAb) (Biolegend, San Diego, CA, USA) as described previously [6].

### 2.9. Statistical Analysis

All statistical analyses were performed using GraphPad Prism version 8 (GraphPad Software, Inc., La Jolla, CA, USA). The mean duration of diarrhea and fecal RV shedding post-challenge were analyzed using an unpaired *t*-test. Log-transformed RV-specific IgA Ab antibodies were analyzed using two-way ANOVA followed by Duncan’s multiple range test. The area under the curve (AUC) analysis was conducted to compare diarrhea severity and shedding among the groups [60]. A Kruskal–Wallis rank sum test was then performed to compare the total AUC values between the groups. Differences were considered significant at *p* ≤ 0.05.

## 3. Results

### 3.1. HBGA-Expressing Bacteria (HBGA^+^) Cocktail Reduces Diarrhea Severity and Virus Shedding

The presence of individual strains of both HBGA^+^ and HBGA^−^ bacterial cocktails in rectal swabs was confirmed by using species-specific primers in PCR analysis. Total aerobic bacterial counts in piglets colonized with HBGA^−^ vs. HBGA^+^ bacteria did not differ, while the numbers of anaerobic bacteria were significantly higher in the piglets colonized with HBGA^−^ bacteria (Appendix A, *p* < 0.05). Diarrhea onset after infection with RVA (G5P[7], G4P[6]), RVC G3P[18], and RVB strains was significantly delayed in the HBGA^+^ vs. HBGA^−^ piglets (Table 1). Further, the RV-induced diarrhea lasted significantly longer in the piglets colonized with HBGA^−^ bacteria, in groups infected with RVA G4P[6] and RVB strains (Table 1, *p* < 0.05). In addition, piglets colonized with HBGA^+^ bacteria displayed a significant reduction in the mean cumulative fecal score (Table 1, *p* < 0.05), in the RVB-infected piglets. A significantly lower AUC value was noted in the piglets colonized with the HBGA^+^ bacterial cocktail after infection with RVA G4P[6], G9P[13], and RVB (Table 1, *p* < 0.05). Further, statistically significant decreases in diarrhea severity were noted at dpi 1 for the RVB-infected piglets (Figure 1E, *p* < 0.05); at dpi 6 for the RVA G9P[13]-infected piglets (Figure 1D, *p* < 0.01), and at dpi 7 for the RVA G4P[6]- and G9P[13]-infected piglets (Figure 1C,D, *p* < 0.05).

Consistent with the clinical data, we observed a significantly delayed onset of virus shedding in the piglets colonized with HBGA^+^ bacteria after infection with RVC strains G6P[5] and G3P[18] (Table 2, *p* < 0.05) compared to the piglets colonized with HBGA^−^ bacteria. In addition, HBGA+-inoculated piglets had a significantly shortened duration of virus shedding in piglets infected with RVA G1P[8] and RVC G6P[5] (Table 2, *p* < 0.05). Significantly lower viral shedding titers in this group were observed on dpi 1 after infection with RVA G4P[6], G9P[13], RVB (Figure 2C–E, *p* < 0.001), and RVC G6P[5] (Figure 2H, *p* < 0.05); on dpi 2 after infection with RVA G4P[6] (Figure 2C, *p* < 0.05); and at dpi 6 after infection with RVA G5P[7] and RVC G1P[1] (Figure 2B,F, *p* < 0.05).

### 3.2. There Was No Evidence That the Protective Effect of HBGA+ Bacteria Was Immune-Mediated

To confirm that the observed protective effect of HBGA^+^ bacteria on RV infections was not associated with bacteria-mediated immunomodulation, we evaluated the local (intestinal content) and systemic (serum) RV-specific IgA Ab responses. Data on the mean RVA/RVC-specific IgA Ab titers in the blood (Figure 3) and intestinal contents (Figure 4) revealed that colonization with HBGA+ bacteria did not result in significantly higher IgA Ab titers (compared to piglets colonized with HBGA^−^ bacteria) after infection with RVA/RVC strains. In addition, significantly lower IgA Ab titers in the blood of HBGA^+^ bacteria-colonized piglets were observed after infection with RVA G1P[8] (Figure 3A, *p* < 0.01), RVA G5P[7] (Figure 3B, *p* < 0.001), and RVC G1P[1] (Figure 3E, *p* < 0.05). This coincided with significantly lower IgA Ab titers in the intestinal contents of the piglets colonized with HBGA^+^ bacteria after infection with RVA G1P[8] and G9P[13] (Figure 4A,D, *p* < 0.01).

We also evaluated the cytokine profiles in blood samples collected before (dpi 0) and after (dpi 3) virus inoculation (Appendix A). In piglets colonized with HBGA^-^ bacteria, we observed a higher IFN-α concentration compared to those colonized with HBGA^+^ bacteria after infection with RVA G5P[7] (Appendix A, *p* < 0.05). The TNF-α concentrations were higher following RVC G3P[18] infection in the piglets colonized with HBGA^+^ vs. HBGA^−^ bacteria (Appendix A, *p* < 0.05). However, the rest of the data on TNF-α and IL-22/IL-10 responses did not allow for discrimination between piglets colonized with HBGA^+^ vs. HBGA^-^ bacterial cocktails.

### 3.3. The Protective Effect of HBGA+ Bacteria Did Not Vary with the Piglet HBGA Phenotype

Although previous studies suggested that the host HBGA phenotype plays an important role in RV infection and evolution [61,62], there are no in vivo data for porcine RVs [6,7]. Here, we aimed to establish whether the protective effect of the bacterial cocktail was independent of the piglet HBGA phenotype. In our study, the data for piglets with the A^strong^ (A^+^H^−^) and A^weak^ (A^+^H^+^) phenotypes were combined and compared with the O phenotype (A^−^H^+^). Our analysis demonstrated that there were no significant differences in diarrhea severity and virus shedding associated with the A^+^ vs. A^−^ phenotypes (Appendix A). This indicates that the protective effect of HBGA^+^ bacteria on RV infection was not affected by piglet HBGA phenotype.

## 4. Discussion

The tripartite RV–host–commensal bacteria interactions have been demonstrated to have profound impacts on RV infection and disease [13,18,63]. In addition to the known mechanisms of protection, such as immunomodulation [64], metabolic and enzymatic support [65], and improved barrier function [66] utilized by probiotics, studies have shown that certain non-pathogenic bacteria possess the ability to directly bind certain viruses [67], including RVs [13,68]. For RVs, this phenomenon has been shown to be associated with the ability of bacteria to express structures similar to what RV uses as attachment sites on IECs, such as HBGAs [16]. However, there is no consensus opinion on the significance of these interactions in vivo [13,69,70]. The current study aimed to evaluate whether direct binding of RV by HBGA^+^ bacteria is associated with reduced or enhanced RV infection and disease.

Our current data suggest that HBGA^+^ bacteria improved protection against RVA, RVB, and RVC infection compared to HBGA^-^ bacteria [16]. Although several studies have shown that the presence of bacterial HBGA-like structures enhanced viral replication [71,72], our study demonstrated that the direct binding of RV virions may represent an additional mechanism of antiviral protection conferred by probiotic/commensal bacteria. These contrasting findings may be attributed to the use of in vitro models (cell culture) in other studies, which lack a key component of the RV–host–bacteria interactions, the mucus, and, thus, may not be physiologically relevant. While RV binding by bacteria can facilitate RV particle delivery to target cells in vitro, the intestinal mucus in the gut may significantly restrict direct contact between bacteria and IECs [73,74], thus limiting the ability of bacteria to serve as a “Trojan horse” for viruses.

There are multiple studies on the use of probiotics against RV infection in which the beneficial/protective effects of certain bacteria are linked to the immunomodulatory effects of bacteria [75,76,77,78]. Our findings indicated that in most cases, the IgA Ab titers and IFN-α concentrations following RV infection were either similar or lower in the piglets colonized with HBGA^+^ bacteria. This suggests that the beneficial effect of HBGA^+^ bacteria was not associated with improved immune responses. In contrast, increased IgA Ab titers and cytokine concentrations in most cases could be associated with increased virus replication in the piglets colonized with HBGA^-^ bacteria.

Our study has several limitations related to the inability to control all the aspects of the bacteria–RV interactions between the two different bacterial cocktails. First, in this study, we utilized a different number of bacterial strains in probiotic cocktails (five in the HBGA^+^ and two in the HBGA^−^). However, our previous study demonstrated that an increased number of probiotics in treatment does not result in superior anti-RV protection compared to an individual probiotic treatment [20]. In our current study, both HBGA^+^ and HBGA^-^ bacteria colonized GF piglets effectively, with higher total bacterial counts observed in piglets colonized with HBGA^-^ bacteria, which could be due to the individual growing characteristics of bacterial strains. This indicates that the higher number of bacterial strains used in the HBGA^+^ bacterial cocktail did not result in a higher bacterial load in the colonized piglets. Next, while we did not have any evidence that the protective effect of the HBGA^+^ probiotic cocktail was immune-mediated, there are other bacteria-mediated effects on the host and RV infection that were not evaluated in this study. For example, bacteria can regulate the mucus composition by stimulating the production or degradation of mucin-type glycans, altering the ability of mucus to provide decoy epitopes for RV attachment. Several probiotics have been shown to up- (*Lactobacilli*) and down-regulate (*Bifidobacteria* and *Streptococci*) mucin secretion [79]. In addition, *B. thetaiotaomicron*, which was used in this study as a component of the HBGA^+^ bacterial cocktail, was previously shown to stimulate mucin secretion in vitro [80]. In addition, various commensal bacteria, including *B. thetaiotaomicron* and *B. longum* were shown to produce sialidases, a group of enzymes responsible for sialic acid removal and mucin degradation, thus affecting the protective role of the intestinal mucus [81,82]. Thus, more studies are needed to dissect the roles of individual/combined HBGA^+^ bacteria in RV infection.

Nevertheless, this is the first study that evaluated a probiotic cocktail broadly protective against genetically distinct RVA, RVB, and RVC strains and demonstrated the role of the HBGA-mediated interactions in this protection in vivo. Thus, our data provide a proof-of-concept that probiotic/commensal bacteria can act as decoy receptors reducing the severity of RV infection and disease in vivo. Further studies are needed to validate the feasibility of this concept in conventional animals.

## 5. Conclusions

In conclusion, we demonstrated the ability of the HBGA^+^ bacteria to reduce species A, B, and C RV infection in vivo. This may represent a novel mechanism for protection against RV-associated diarrhea. However, the protective effects of the HBGA^+^ bacteria in conventional animals, where HBGA^+^ bacteria would need to compete with already established microbiota, remain to be evaluated. In addition, other aspects of RV–host–bacteria interactions, such as enzymatic and metabolic alterations associated with the HBGA^+^ bacterial cocktail and their effects on IEC integrity, must be further investigated.

## Figures and Tables

**Figure 1 viruses-16-00660-f001:**
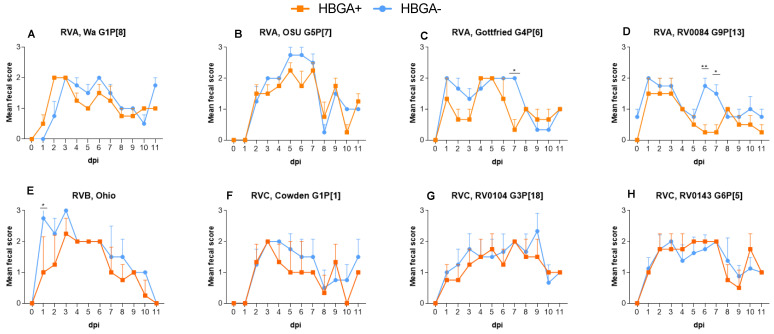
Diarrhea in piglets following RVA (**A**–**D**), RVB (**E**), and RVC (**F**–**H**) inoculation. Individual RV strains were used to inoculate (1 × 10^6^ FFU) piglets after 5 consecutive days of supplementation with HBGA^+^ or HBGA^-^ bacteria. Fecal consistency was scored as follows: 0, normal; 1, pasty; 2, semiliquid; 3, liquid; and diarrhea was considered as a score of ≥2. The error bars represent the standard deviations; significant differences (* *p* < 0.05, ** *p* < 0.01) are indicated as calculated by two-way ANOVA followed by Duncan’s multiple comparisons test.

**Figure 2 viruses-16-00660-f002:**
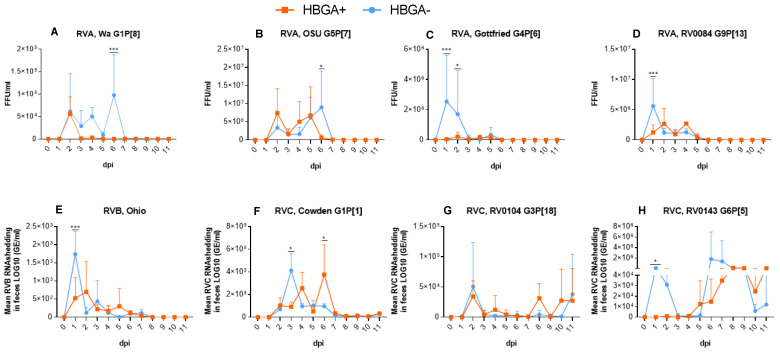
Mean virus shedding titers in piglets following RVA (**A**–**D**), RVB (**E**), and RVC (**F**–**H**) inoculation. Individual RV strains were used to colonize (1 × 10^6^ FFU) piglets after 5 consecutive days of supplementation with HBGA^+^ or HBGA^−^ bacteria. The error bars represent the standard deviations; significant differences (* *p* < 0.05, *** *p* < 0.001) are indicated as calculated by two-way ANOVA followed by Duncan’s multiple comparisons test.

**Figure 3 viruses-16-00660-f003:**
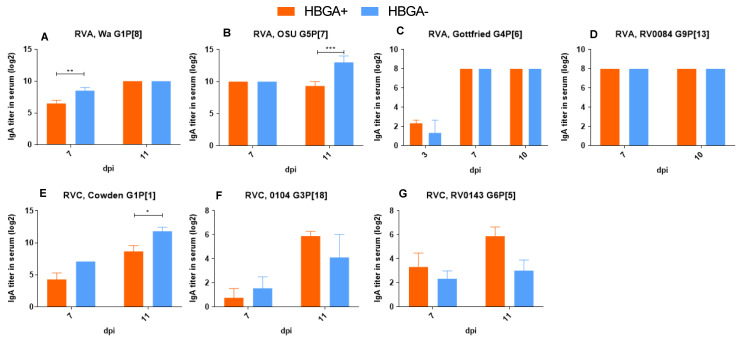
IgA Ab titers in blood samples collected on dpi 0, 3, 7, 11 following RVA (**A**–**D**), and RVC (**E**–**G**) infection. ELISA IgA Ab titers were analyzed using two-way ANOVA followed by Duncan’s multiple comparisons test (* *p* < 0.05, ** *p* < 0.01, *** *p* < 0.001).

**Figure 4 viruses-16-00660-f004:**
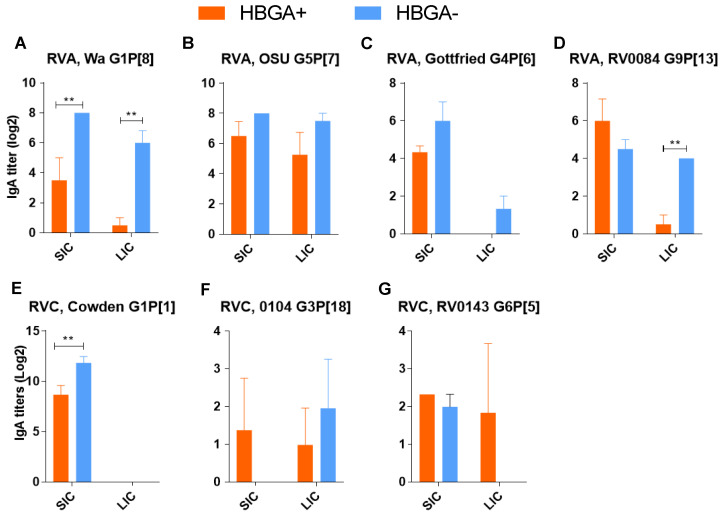
IgA Ab titers in piglet small/large intestinal content (SIC/LIC) samples collected at necropsy 11 days after RVA (**A**–**D**), and RVC (**E**–**G**) infection. ELISA IgG Ab titers were analyzed using two-way ANOVA followed by Duncan’s multiple comparisons test (** *p* < 0.01).

**Table 1 viruses-16-00660-t001:** Diarrhea in piglets orally inoculated with virulent RVs.

	N		Mean Days to Diarrhea Onset ^1^		Mean Diarrhea Duration (Days) ^2^		Mean Cumulative Fecal Score ^3^		AUC ^4^	
RV Strains	HBGA^+^	HBGA^−^	HBGA^+^	HBGA^−^	HBGA^+^	HBGA^−^	HBGA^+^	HBGA^−^	HBGA^+^	HBGA^−^
Wa G1P[8]	4	4	2.0	2.8	3.3	4.8	13.0	13.8	12.50	12.88
OSU G5P[7]	4	4	3.3	2.5	5.5	6.0	15.0	17.0	14.38	16.50
Gottfried G4P[6]	3	3	2.0	1.0	**3.3**	**5.7**	11.7	15.3	**11.17**	**14.83**
RV0084 G9P[13]	4	4	1.0	1.0	2.3	3.8	8.8	13.8	**8.625**	**13.75**
RVB Ohio	4	4	2.0	1.0	**5.3**	**7.3**	**13.5**	**19.0**	**13.50**	**19.00**
Cowden G1P[1]	4	4	2.7	2.8	2.7	4.5	10.3	13.5	9.833	12.75
RV0104 G3P[18]	4	4	4.5	3.8	3.8	5.3	14.3	16.3	13.75	15.83
RV0143 G6P[5]	4	8	2.3	2.1	6.0	5.3	16.3	16.0	15.75	15.50

^1^ Diarrhea onset is defined as the number of days between the virus inoculation and the first manifestation of diarrhea (e.g., fecal consistency score of ≥2). ^2^ Duration of diarrhea is defined as the number of days that the fecal consistency score was ≥2. Fecal diarrhea was scored as follows: 0, normal; 1, pasty; 2, semiliquid; 3, liquid. ^3^ Mean cumulative fecal score [(sum of fecal consistency score for 11 days postinoculation)/N], where N is the number of pigs receiving the inoculation. Means in the same row were analyzed by unpaired *t*-test. ^4^ Area under the curve (AUC) was calculated using the area under the curve analysis function in the Prism software. A Kruskal–Wallis rank sum test was then performed to compare the total AUC between the groups. Significant differences (bold) are indicated as calculated by an unpaired *t*-test.

**Table 2 viruses-16-00660-t002:** Virus shedding in piglets orally inoculated with virulent RVs.

	N		Mean Days to Shedding Onset		Mean Shedding Duration (Days)		Avg Peak Titer (FFU/mL)		AUC ^1^	
RV Strains	HBGA^+^	HBGA^−^	HBGA^+^	HBGA^−^	HBGA^+^	HBGA^−^	HBGA^+^	HBGA^−^	HBGA^+^	HBGA^−^
Wa G1P[8]	4	4	2.0	2.0	**6.3**	**8.5**	5.58 × 10^4^	6.14 × 10^4^	6.04 × 10^4^	2.50 × 10^5^
OSU G5P[7]	4	4	2.0	2.0	6.8	6.5	7.41 × 10^6^	9.01 × 10^6^	2.16 × 10^7^	2.18 × 10^7^
Gottfried G4P[6]	3	3	2.0	1.7	4.7	5.7	6.63 × 10^5^	4.83 × 10^6^	6.62 × 10^5^	4.82 × 10^6^
RV0084 G9P[13]	4	4	1.3	1.0	7.5	8.3	8.02 × 10^6^	9.85 × 10^6^	8.04 × 10^6^	9.87 × 10^6^
RVB Ohio	4	4	1.0	1.0	7.0	7.0	7.00 × 10^2^	1.74 × 10^3^	2.07 × 10^3^	2.64 × 10^3^
Cowden G1P[1]	4	4	2.0	2.0	10.0	10.0	3.76 × 10^5^	4.12 × 10^5^	9.60 × 10^5^	8.29 × 10^5^
RV0104 G3P[18]	4	4	**2.0**	**1.0**	9.8	10.3	5.07 × 10^4^	3.94 × 10^4^	1.32 × 10^5^	9.23 × 10^4^
RV0143 G6P[5]	4	8	**3.5**	**1.1**	**8.0**	**10.5**	1.60 × 10^5^	1.83 × 10^6^	3.97 × 10^5^	3.53 × 10^6^

Means in the same row were analyzed by unpaired *t*-test. ^1^ Area under the curve (AUC) was calculated using the area under the curve analysis function in the Prism software. A Kruskal–Wallis rank sum test was then performed to compare the total AUC between the groups. Significant differences (bold) are indicated as calculated by an unpaired *t*-test.

## Data Availability

Data are contained within the article.

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
