# Peer review of "Histo-Blood Group Antigen-Producing Bacterial Cocktail Reduces Rotavirus A, B, and C Infection and Disease in Gnotobiotic Piglets"

_viruses, 2024, doi:10.3390/v16050660_

Round 1
Reviewer 1 Report
Comments and Suggestions for Authors
In this manuscript, authors demonstrated the ability of the HBGA+ bacteria to reduce species A, B and C RV infection in vivo. They found that HBGA+ cocktail reduces diarrhea severity and virus shedding, and the beneficial effect of HBGA+ bacteria was not associated with improved immune responses. The topic of this manuscript is of interest for the field; however, some data are not convincing, and additional experiments should be conducted to support their conclusions.
1. We know that in experiments involving pigs, there are usually at least 5 pigs in each group. In this animal experiment, how many pigs are used in each group?
2. How do you ensure that bacteria have colonized the intestines of piglets before virus infection, and how do you measure the number of bacteria in the piglet's gut?
3. HBGA+ Cocktail Reduces Diarrhea Severity and Virus Shedding. After RV infection, do bacteria have a certain effect on viremia in piglets?
Comments on the Quality of English LanguageModerate editing of English language required
Author Response
We sincerely thank the Reviewer for the constructive criticism, insightful comments, and valuable feedback on our manuscript. Please see our point-by-point responses below.
- In this manuscript, authors demonstrated the ability of the HBGA+ bacteria to reduce species A, B and C RV infection in vivo. They found that HBGA+ cocktail reduces diarrhea severity and virus shedding, and the beneficial effect of HBGA+ bacteria was not associated with improved immune responses. The topic of this manuscript is of interest for the field; however, some data are not convincing, and additional experiments should be conducted to support their conclusions.
AU: We thank the reviewer for the overall positive evaluation of our manuscript. This study is a continuation of our previous in vitro study, where we evaluated the ability of individual bacterial strains to directly bind RV species. The current study demonstrated that HBGA-expressing bacteria can reduce diarrhea severity/viral shedding to a greater extent than lactobacilli and bifidobacteria that do not express HBGA but are commonly used as probiotics.
- We know that in experiments involving pigs, there are usually at least 5 pigs in each group. In this animal experiment, how many pigs are used in each group?
AU: Considering the number of different treatments, we aimed to use 4 piglets/group, because N=3-4 is the minimum allowing to conduct statistical analysis. We had it achieved for all of the RV groups but one – Gottfried. We were limited by several factors, including logistical constraints (pig litter sizes), ethical considerations, and the specific objectives of our study.
- How do you ensure that bacteria have colonized the intestines of piglets before virus infection, and how do you measure the number of bacteria in the piglet's gut?
AU: As was described in our manuscript, on day 5 of supplementation, rectal swabs were collected to enumerate total aerobic and anaerobic bacterial counts. We have added a relevant reference. Additionally, the presence of individual strains of both HBGA+ and HBGA- bacterial cocktails in rectal swabs was confirmed using species-specific primers in PCR analysis (primers are listed in the supplementary file).
- HBGA+ Cocktail Reduces Diarrhea Severity and Virus Shedding. After RV infection, do bacteria have a certain effect on viremia in piglets?
AU: RV viremia is very transient and not consistently detected in all of the piglets. Moreover, its presence or absence does not seem to correlate with the disease severity. Thus, we did not evaluate the effect of bacteria on viremia.
Reviewer 2 Report
Comments and Suggestions for Authors
In the ms entitled “Histo-Blood Group Antigen-Producing Bacteria Cocktail Reduces Rotavirus A, B and C Infection and Disease in Gnotobiotic Piglets”, Raev et al. showed that colonization with HBGA+ bacteria resulted in reduced diarrhea severity and Rotavirus shedding compared to the HBGA- bacteria in an immune responses-independent manner. Additionally, these beneficial effects were not related to the piglet’s HBGA phenotype. These findings provide in vivo evidence for HBGA+ bacteria to provide protection piglets from Rotavirus via decoy epitopes. The authors undertook an interesting work, and the results provides more insight for the beneficial effects of probiotics on the intestinal health of piglets.
Some comments are list below:
1. Line 163-165, “Log-transformed RV-specific IgA Ab antibodies were compared using were analyzed using two-way ANOVA followed by Duncans multiple range test.”, Please check the grammar of this sentence.
2. Figure 2, symbols for HBGA+ and HBGA- should be shown as in Figure 1, not just in C.
3. In supplementary materials, the first letter of the title on the vertical axis of Figure 1 should be capitalized.
4. In notes of Supplementary Table 2, “2Duration of diarrhea is defined as the number of days that the diarrhea score was 2.”, Why is it that only the days with a diarrhea score of 2 points are counted here, and not the days with a score exceeding 2 points? And the Significant differences were not marked in the table.
Author Response
We sincerely thank the Reviewer for the constructive criticism, insightful comments, and valuable feedback on our manuscript. Please see our point-by-point responses below.
- In the ms entitled “Histo-Blood Group Antigen-Producing Bacteria Cocktail Reduces Rotavirus A, B and C Infection and Disease in Gnotobiotic Piglets”, Raev et al. showed that colonization with HBGA+ bacteria resulted in reduced diarrhea severity and Rotavirus shedding compared to the HBGA- bacteria in an immune responses-independent manner. Additionally, these beneficial effects were not related to the piglet’s HBGA phenotype. These findings provide in vivo evidence for HBGA+ bacteria to provide protection piglets from Rotavirus via decoy epitopes. The authors undertook an interesting work, and the results provide more insight for the beneficial effects of probiotics on the intestinal health of piglets.
AU: We thank the reviewer for the overall positive evaluation of our manuscript.
- Line 163-165, “Log-transformed RV-specific IgA Ab antibodies were compared using were analyzed using two-way ANOVA followed by Duncans multiple range test.”, Please check the grammar of this sentence.
AU: Thank you for catching this. We have revised the sentence.
- Figure 2, symbols for HBGA+ and HBGA- should be shown as in Figure 1, not just in C.
AU: Revised as suggested
- In supplementary materials, the first letter of the title on the vertical axis of Figure 1 should be capitalized.
AU: Revised as suggested
- In notes of Supplementary Table 2, “2Duration of diarrhea is defined as the number of days that the diarrhea score was 2.”, Why is it that only the days with a diarrhea score of 2 points are counted here, and not the days with a score exceeding 2 points?
AU: We apologize for that. Revised.
- And the Significant differences were not marked in the table.
AU: We apologize for the copy-paste error. There were no significant differences in the data presented in the table.
Reviewer 3 Report
Comments and Suggestions for Authors
Authors tried to demonstrate that HBGA+ expressing bacteria, serving as probiotics, is beneficial in reducing diarrhea score and viral shedding time graph, after gnotobiotic pigs were challenged with rotavirus group A, B, and C. A lots of experiments were done, but the significant data are listed in Table 1 (in particular left half). Most questions (section 3.2 and 3.3) asked resulted in negative findings.
Although authors like to portray that the beneficial effects of this probiotics were immune-mediated or related to HBGA phenotype of pigs, and turned out disappointed. I would rather seeing a probiotics work "locally in the intestine", i.e., competing for the cell surface receptor with RV before viruses get into the cells and disregard the pig blood types. So, I view negative findings in sections 3.2 and 3.3 as strengths (rather than limitations, as discussed in lines 316-322) if these HBGA+ bacteria were to be applied in the field. However, this may contradict with the RV specific IgA test results (authors can think about this further, the only thing matter should be non-specific IgA in the intestinal mucosa). If probiotics are to work systemically in an antigen-specific manner, it is equal to a vaccine. So the best way to use this probiotics is as a complement to the now marketed suboptimal rotavirus vaccine.
minor comments:
I suggest using superscript for the "+ or -" in the HBGA+ or -.
line 46, spell out what is IEC?
section 2.5 and 2.6: indicate here you used samples, both intestinal contents and serum. In immunology we were taught that IgA matters in mucosa, you can explain why serum is included here.
lines 201 to 216; and Table 1: the statements sound very promising, but in Table 1 (last column, AUC) the letters are not bald (as significant), please confirm.
Author Response
We sincerely thank the Reviewer for the constructive criticism, insightful comments, and valuable feedback on our manuscript. Please see our point-by-point responses below.
- Authors tried to demonstrate that HBGA+ expressing bacteria, serving as probiotics, is beneficial in reducing diarrhea score and viral shedding time graph, after gnotobiotic pigs were challenged with rotavirus group A, B, and C. A lots of experiments were done, but the significant data are listed in Table 1 (in particular left half). Most questions (section 3.2 and 3.3) asked resulted in negative findings.
AU: We apologize if we did not clearly stated the study objective, and wanted to mention that our findings, in fact, have confirmed our original hypothesis. In our study, we aimed to demonstrate that the ability of HBGA-expressing commensal bacteria to directly bind RV will provide a beneficial effect protecting against RV infection and disease in vivo. Thus, while our results (section 3.2) indicated that IgA antibody titers and cytokine concentrations following RV infection were either similar or lower in the piglets colonized with HBGA+ bacteria, we do not consider these findings as negative. Instead, they suggest that the beneficial effects of HBGA+ bacteria were not associated with improved immune responses, indicating that the proposed mechanism (direct binding of RV particles by HBGA+ bacteria) played a pivotal role in protection against RV infection.
Previous studies emphasize the role of blood group phenotype in RV replication in vivo. Our findings in section 3.2 revealed that there were no significant differences in RV infection and disease between piglets with different HBGA phenotypes, indicating that the protective effect of HBGA+ bacteria were consistent regardless of piglet blood group. Taken together, both findings from sections 3.2 and 3.3 are consistent with our expectations and are not considered negative. We clarified it throughout.
- Although authors like to portray that the beneficial effects of this probiotics were immune-mediated or related to HBGA phenotype of pigs, and turned out disappointed. I would rather seeing a probiotics work "locally in the intestine", i.e., competing for the cell surface receptor with RV before viruses get into the cells and disregard the pig blood types. So, I view negative findings in sections 3.2 and 3.3 as strengths (rather than limitations, as discussed in lines 316-322) if these HBGA+ bacteria were to be applied in the field. However, this may contradict with the RV specific IgA test results (authors can think about this further, the only thing matter should be non-specific IgA in the intestinal mucosa).
AU: As mentioned in our response to comment #1, we did not anticipate HBGA+ bacteria to confer beneficial effects through immunomodulation or pig HBGA phenotype. Rather, we assessed these parameters to eliminate them as significant contributors to the protective effect of the HBGA+ bacterial cocktail. Similarly, we did not aim to investigate bacterial lectin-mediated effects such as blocking glycans (RV attachment sites) on intestinal epithelial cells.
- If probiotics are to work systemically in an antigen-specific manner, it is equal to a vaccine. So the best way to use this probiotics is as a complement to the now marketed suboptimal rotavirus vaccine.
AU: We appreciate the reviewer's suggestion. Studying the combined vaccine/probiotic treatment's effect on RV infection would indeed be intriguing. However, while vaccines aim to stimulate the immune response for protection, our study evaluated the effect of the ability of HBGA+ bacteria to directly bind RVs. Additionally, there is no compelling data demonstrating that such a ‘universal vaccine’ is feasible.
minor comments:
- I suggest using superscript for the "+ or -" in the HBGA+ or -.
AU: Revised as suggested.
- line 46, spell out what is IEC?
AU: We spelled it out in line 33 as follows: … intestinal epithelial cells (IECs)…
- section 2.5 and 2.6: indicate here you used samples, both intestinal contents and serum. In immunology we were taught that IgA matters in mucosa, you can explain why serum is included here.
AU: Previous studies have shown that IgA antibody titers in serum were highly correlated P < 0.001) with the corresponding isotype antibody (IgA) titers in the gut indicating that the IgA antibody titer is probably the most reliable indicator of protection. (Azevedo MS, Yuan L, Iosef C, Chang KO, Kim Y, Nguyen TV, Saif LJ. Magnitude of serum and intestinal antibody responses induced by sequential replicating and nonreplicating rotavirus vaccines in gnotobiotic pigs and correlation with protection. Clin Diagn Lab Immunol. 2004 Jan;11(1):12-20. doi: 10.1128/cdli.11.1.12-20.2004. PMID: 14715539; PMCID: PMC321356.). Moreover, this topic has been evaluated in another paper also indicating the possible importance of serum IgA in protection against RV disease. (Jiang B, Gentsch JR, Glass RI. The role of serum antibodies in the protection against rotavirus disease: an overview. Clin Infect Dis. 2002 May 15;34(10):1351-61. doi: 10.1086/340103. Epub 2002 Apr 22. PMID: 11981731.)Despite other findings suggested that IgA has a minor role in clearing a primary rotavirus infection from systemic circulation and is not required for protection against rotavirus antigenemia upon re-exposure to the virus [Blutt, S., Miller, A., Salmon, S. et al. IgA is important for clearance and critical for protection from rotavirus infection. Mucosal Immunol 5, 712–719 (2012)], we still believe it is relevant assessing both local (intestinal contents) and systemic (serum) IgA antibody titers for the current and future studies.
Additionally, serum IgA antibody levels are higher and are characterized by a broader dynamic rage than those in intestinal contents. Thus, we included the data on both serum and intestinal IgA antibody levels.
- lines 201 to 216; and Table 1: the statements sound very promising, but in Table 1 (last column, AUC) the letters are not bald(as significant), please confirm.
AU: Our analysis revealed that the protective effect of HBGA+ bacteria was RV species/strain specific. In lines 201-216 we focused on statistically significant results. Thus, while for most RV strains used in this study, colonization of piglets with HBGA+ bacteria resulted in marginal decreases of AUC, these differences were not statistically significant.
Round 2
Reviewer 1 Report
Comments and Suggestions for Authors
NO
Reviewer 3 Report
Comments and Suggestions for Authors
The R1 version has improved. Some data in the original version has been moved to the supplementary making it more smooth to read.